# Polarimetric SAR Speckle Filtering Using a Nonlocal Weighted LMMSE Filter

**DOI:** 10.3390/s21217393

**Published:** 2021-11-06

**Authors:** Yinbin Shen, Xiaoshuang Ma, Shengyuan Zhu, Jiangong Xu

**Affiliations:** 1China JIKAN Research Institute of Engineering Investigations and Design, Co., Ltd., Xi’an 710000, China; yinbinshen@126.com (Y.S.); tarenjidiyu@163.com (S.Z.); 2School of Resources and Environmental Engineering/Anhui Province Key Laboratory of Wetland Ecosystem Protection and Restoration, Anhui University, Hefei 230601, China; x19201019@stu.ahu.edu.cn

**Keywords:** polarimetric synthetic aperture radar, speckle filtering, complex Wishart distribution, nonlocal means

## Abstract

Despeckling is a key preprocessing step for applications using PolSAR data in most cases. In this paper, a technique based on a nonlocal weighted linear minimum mean-squared error (NWLMMSE) filter is proposed for polarimetric synthetic aperture radar (PolSAR) speckle filtering. In the process of filtering a pixel by the LMMSE estimator, the idea of nonlocal means is employed to evaluate the weights of the samples in the estimator, based on the statistical equalities between the neighborhoods of the sample pixels and the processed pixel. The NWLMMSE estimator is then derived. In the preliminary processing, an effective step is taken to preclassify the pixels, aiming at preserving point targets and considering the similarity of the scattering mechanisms between pixels in the subsequent filter. A simulated image and two real-world PolSAR images are used for illustration, and the experiments show that this filter is effective in speckle reduction, while effectively preserving strong point targets, edges, and the polarimetric scattering mechanism.

## 1. Introduction

It is well known that polarimetric synthetic aperture radar (PolSAR) images are inherently affected by speckle noise, which is due to the coherent interference of radar signals reflected from many tiny scatterers in a resolution unit. Speckle noise has long been recognized as one of the most crucial problems of SAR data, and its presence degrades the appearance of images and has a great impact on the performance of land-use classification and scene analysis tasks [1].

One of the earliest methods for SAR image despeckling was presented by Novak and Burl [2], in which the polarimetric whitening filter (PWF) was utilized to produce a single speckle-reduced intensity image by optimally combining all the elements of the polarimetric covariance matrix. Motivated by this method, Lopes and Sery [3] and Liu et al. [4] introduced a PWF for multi-look processed data (MPWF). The above filters exploit the statistical correlations between the different polarization channels, leading to the problem that the polarimetric properties are not carefully preserved. An important branch of filtering methods for PolSAR data is based on the linear minimum mean-squared error (LMMSE) estimator. Lee et al. expanded their local LMMSE (LLMMSE) method in [5] to filter PolSAR data by developing the edge-aligned window technique [1]. In [6], a novel method was further proposed to select neighboring pixels based on the same scattering characteristics, and to use the selected pixels to filter the processed pixel based on the LLMMSE filter. More LLMMSE algorithms can also be found in [7,8,9].

In the traditional methods based on the LLMMSE filter, to obtain more precise filter parameters in the LMMSE estimator, a group of homogeneous image pixels are first selected in a local window. The LMMSE estimator is then generated from the values of the selected pixels, and is saved as the filtered value of the pixel being processed. These methods assume that all of the selected pixels are highly homogeneous pixels with respect to the processed pixel, which is not reliable enough. Besides, most of these methods have the shortcomings of the limited selection range of pixels, and comparing pixels only by their own characteristics, which may produce a biased or inferior estimation of the filter parameters in the estimator.

In addition to the methods based on LMMSE, some other filtering algorithms, such as partial differential equation (PDE)-based filters [10,11], variational-based filters [12,13,14], simultaneous sparse coding (SSC)-based filters [15], spherically invariant random vector-based filters [16], and Bayesian theory-based filters [17] have also been developed in recent years. What must be mentioned is that nonlocal means (NL-means) [18], which were first used in the processing of optical images, have recently been extended for SAR despeckling and have achieved very positive performances. In [19,20], the NL-means were limited in processing single-polarization amplitude images. In [21], the methodology of NL-means was innovatively applied to PolSAR speckle reduction by Chen et al., and they successfully made use of the full polarimetric information. In [13], the ideas of total variation (TV) and NL-means were combined to tackle the issue of PolSAR image despeckling. In the NL-means method, the restored value of each pixel is obtained by the weighted average of all the pixels in a large searching window. Each weight is proportional to the equality between the local neighborhood of the pixel being processed and the neighborhood of the other pixels. The basic idea is that images contain repeated structures, and averaging them will reduce the random noise. This provides us with a new thought to solve the problems that the conventional LLMMSE filters encountered in filtering PolSAR images: it is better to assign a weight or reliability to each pixel in the LMMSE estimator based on its equality of the full polarimetric information with the processed pixel, rather than regard it as an absolutely homogeneous pixel. By doing so, the selection window of pixels can also be expanded, which can ensure that enough similar samples are included in the filter; in addition, the weight can be obtained by comparing the equality of the pixels’ neighborhoods, rather than only comparing the value in a single point.

Based on the above viewpoints, a novel nonlocal weighted linear minimum mean-squared error (NWLMMSE) filter is investigated in this present research, aiming at solving the problems of the conventional PolSAR LLMMSE filters and taking advantage of the theory of NL-means. First, an effective preliminary step is taken to preclassify the pixels. Then, strong point targets detected in the above step are kept unfiltered; for distributed target pixels, only the similar pixels in the searching window are selected as samples, and the idea of NL-means is employed to calculate the weights of the selected pixels. Finally, the weighted filter parameters evaluated by these pixels are applied to the LMMSE estimator, and the NWLMMSE filter is derived to filter the processed pixel.

The remainder of this paper is organized as follows. First, some basic concepts of the local linear minimum mean-squared error filter and NL-means are introduced in Section 2, and the proposed NWLMMSE method is described in Section 3. Then, in Section 4, the experimental results and analyses of a simulated and two real-world PolSAR images are given. Finally, the conclusions are drawn in Section 5.

## 2. The LLMMSE Filter and Nonlocal Means

### 2.1. The LLMMSE Filter

PolSAR data can be represented in the following polarimetric covariance matrix:(1)$$C=\left[\begin{array}{ccc} |S_{HH}{|}^{2} & \sqrt{2}\left|S_{HH}\right|\left|S_{HV}\right|e^{j(\Phi_{HH}-\Phi_{HV)}} & \left|S_{HH}\right|\left|S_{VV}\right|e^{j(\Phi_{HH}-\Phi_{VV)}}\\ \sqrt{2}\left|S_{HH}\right|\left|S_{HV}\right|e^{j(\Phi_{HV}-\Phi_{HH)}} & 2|S_{HV}{|}^{2} & \sqrt{2}\left|S_{VV}\right|\left|S_{HV}\right|e^{j(\Phi_{HV}-\Phi_{VV)}}\\ \left|S_{HH}\right|\left|S_{VV}\right|e^{j(\Phi_{VV}-\Phi_{HH)}} & \sqrt{2}\left|S_{VV}\right|\left|S_{HV}\right|e^{j(\Phi_{VV}-\Phi_{HV)}} & |S_{VV}{|}^{2} \end{array}\right]$$
where $S_{HV}$ denotes the scattering element of the horizontal receiving and vertical transmitting polarizations, with the combination of the amplitude $\left|S_{HV}\right|$ and the phase $\phi_{HV}$: $S_{HV}=\left|S_{HV}\right|e^{j\phi_{HV}}$, and *j* denotes the imaginary part. For fully developed speckle, multi-look covariance matrix follows a complex Wishart distribution [22].

The speckle in a SAR intensity image is described in terms of a multiplicative noise model:(2)y=xν
where x is the noise-free pixel value to be estimated, and v is the noise with mean 1 and variance $\sigma_{v}{}^{2}$. $\sigma_{v}$ is a measure of the speckle level and has been shown to be the standard-to-mean ratio for homogeneous areas [23].

Considering $\bar{x}$ as an a priori mean of x, the LLMMSE filter is developed by a linear combination of $\bar{x}$ and y, which is
(3)$$x=a\bar{x}+by$$
where $\bar{x}$ is evaluated by $\bar{y}$, the local mean of y. The goal of this filter is to minimize the mean square error (MSE):(4)$$J=E[\left(x-\bar{x}{)}^{2}\right].$$

By optimally choosing parameters a and b to minimize the MSE, the LMMSE estimator can be obtained as follows [5]:(5)$$x=\bar{y}+b\left(y-\bar{y}\right),$$
with
(6)$$b=\frac{\mathrm{var}(x)}{\mathrm{var}(y)}=\frac{\mathrm{var}(y)-{\bar{y}}^{2}*\sigma_{v}^{2}}{\left(1+\sigma_{v}^{2}\right)*\mathrm{var}(y)}.$$

The parameter b works as a weight between the local mean and the original pixel value. For homogeneous areas, var(x)≈0, and $x=\bar{y}$, the local mean value. For heterogeneous areas with highly complicated image structures, var(x)≈var(y), and x=y, the original value of the center pixel.

Lee et al. [1] first applied the LMMSE estimator to filter the whole covariance matrix of PolSAR data, including the off-diagonal terms. The filtered covariance matrix is
(7)$$C=\bar{C}+b\left(C-\bar{C}\right),$$
where C is the covariance matrix of the processed pixel, and $\bar{C}$ is the local mean of the covariance matrix. To preserve the polarimetric properties and avoid crosstalk between polarization channels, the same weight b, which is computed in the span image, is used to filter each term of the covariance matrix independently. Clearly, in the LMMSE estimator, the estimation of the a priori mean and the coefficient of variation determines its performance in speckle reduction. A more accurate evaluation of these parameters is essential. Hence, the conventional LLMMSE methods have made great efforts to accurately estimate the filter parameters. The general approach is to select the homogeneous pixels in the local neighborhood of the processed pixel, and then calculate the local mean and coefficient of variation by these homogeneous pixels. In view of estimation theory, we believe that the conventional LLMMSE methods have three main problems:In the conventional LLMMSE filters, the filter parameters are evaluated by the selected pixels (samples), with the assumption that all of the samples are absolutely homogeneous pixels with respect to the processed pixel. However, this is not generally true. Due to the spatial variations of the scene signal, the measured radar signal is not generally stationary, and the estimation of the filter parameters (e.g., the mean and the coefficient of variation) leads to meaningless values. Non-stationarity might be due to the presence of edges, curvilinear features, or point targets. Scene signals might be non-stationary even within a small region [24].To obtain homogeneous pixels, various scene templates proposed in the literature have been used to match the best local stationarity [1], [6]. After an optimum matching of the stationary features, speckle filtering is adapted to these matched features. However, being subject to their algorithms, most of these LLMMSE methods are limited to a small local window (often in a 7 × 7 or 9 × 9 window) when selecting the homogeneous pixels, which may bring about inferior estimation on filter parameters, because a large processing window is generally needed for accurate estimation of statistics at the deepest level. A bias might be introduced if the number of independent samples is not sufficiently large.When selecting the homogeneous pixels, only their own characteristics are considered, which is not robust enough, as the information of their neighborhoods could also be taken into account.

### 2.2. Nonlocal Means

Buades et al. [16] first introduced NL-means for digital image denoising, taking advantage of similar patches in the same image. Since then, NL-means based algorithms have been extensively studied for various applications [25,26,27], and achieved state-of-the-art performances. In NL-means, the restored value of each pixel is obtained by the weighted average of all the pixels in a large searching window. Each weight is proportional to the equality between the local neighborhood of the pixel being processed and the neighborhood of the other pixels. The NL-means not only compare the values at a single point but the geometrical configuration in a whole neighborhood, which allows for a more robust comparison than local filters.

The NL-means technique was firstly applied to PolSAR speckle reduction by Chen et al. [21]. The similarity of two pixels in that paper was defined by a likelihood-ratio test statistic for the equality of two covariance matrices based on the complex Wishart distribution. By comparing the similarity of two patches, the equality of corresponding pixels located at the centers was obtained. An empirical threshold of the likelihood-ratio test was given. Only those pixels that satisfied the similarity test were considered as “homogeneous pixels”, and the equality between two homogeneous pixels was represented by a weight function, which induced a weighted averaging filter. In this study, we also employ the idea of NL-means to calculate the weights of pixels. Compared with the scheme of NL-means proposed in [21], our contributions are as follows:We believe that two pixels are similar in scattering property if they are similar in both statistical property and polarimetric scattering mechanism. As pointed out in [28], if only a statistical criterion is applied, pixels dominated by surface scattering could be considered statistically similar to pixels dominated by volume scattering. Therefore, in this study, the polarimetric scattering mechanism is also considered to measure the similarity between pixels.Preservation of strong returns from point targets is essential for target and man-made structure detection. Unlike scattering from distributed media or targets, scattering from point targets comes mainly from a few strong scatterers within a resolution cell, and these targets do not possess the typical characteristics of speckle. It is better to keep them unfiltered or filter them differently from the distributed targets. In this present research, we take an effective step to detect strong point targets and retain their original values.High computational complexity is the main problem of NL-means, which makes this method quite slow in practice. The high computational complexity is due to the cost of the equality calculation for all the pixels in a large window. In this paper, we make some further efforts to ease the computational load.More importantly**,** we combine the LMMSE estimator and NL-means to reduce PolSAR speckle, which not only makes full use of the advantages of NL-means, but also takes account of the property of the multiplicative noise.

## 3. The Proposed Nonlocal Weighted LMMSE Filter

The basic idea of our proposed nonlocal weighted LMMSE filter is that, to obtain a precise estimation of the filter parameters of the LMMSE estimator, enough statistical samples should be selected without considering whether they are absolutely homogeneous pixels or not (i.e., loosening the constraint of selecting samples), and a weight or reliability should be assigned to each sample. This can be described as the following statistical model: suppose we have n samples. Their covariance matrices are $C_{1},$ $C_{2},$ …, $C_{n}$, respectively, and their weights are $w_{1},$ $w_{2},$ …, $w_{n}$, respectively. The span values are then, respectively, $S_{1},$ $S_{2},$ …, $S_{n}$. Assume that the estimated value is a linear combination of a priori mean $\bar{x}$ and y, and following the process of the LMMSE estimator, the filtered covariance matrix of the processed pixel is estimated by the NWLMMSE filter as
(8)$$\begin{array}{l} C={\bar{C}}_{w}+b*\left(C-{\bar{C}}_{w}\right)\\ =\left(1-b\right)*{\bar{C}}_{w}+b*C \end{array}$$
where ${\bar{C}}_{w}$ is the weighted mean value of the samples’ covariance matrices:(9)$${\bar{C}}_{w}=W_{1}*C_{1}+W_{2}*C_{2}+\cdots +W_{n}*C_{n}\;.$$

The span image is used in the computation of the weight b:(10)$$b=\frac{\mathrm{var}{\left(y\right)}_{w}-{\bar{y}}_{w}*\sigma_{v}{}^{2}}{\left(1+\sigma_{v}{}^{2}\right)*\mathrm{var}{\left(y\right)}_{w}}\;$$
where the mean value ${\bar{y}}_{w}$ and variance $\mathrm{var}{\left(y\right)}_{w}$ are also obtained by considering the weight of each sample as the following Equation:(11)$${\bar{y}}_{w}=W_{1}*S_{1}+W_{2}*S_{2}+\cdots +W_{n}*S_{n\;}$$
(12)$$\mathrm{var}{\left(y\right)}_{w}=W_{1}*{\left(S_{1}-{\bar{y}}_{w}\right)}^{2}+W_{2}*{\left(S_{2}-{\bar{y}}_{w}\right)}^{2}+\cdots +W_{n}*{\left(S_{n}-{\bar{y}}_{w}\right)}^{2}.$$

In NL-means, the estimated value is considered as the weighted mean value of samples: $C={\bar{C}}_{w}$; in the NWLMMSE filter, it is assumed to be a linear combination of the weighted mean value ${\bar{C}}_{w}$ and the processed pixel value C: $C=\left(1-b\right){\bar{C}}_{w}+bC$. The parameter b works as a weight between the weighted local mean value and the original pixel value. We can see that the formulation of NL-means is just a special case of the NWLMMSE estimator when b=0. This indicates that the NWLMMSE filter is adaptive, and not only employs the idea of NL-means, but also takes account of the property of the multiplicative noise.

The main question with regard to the NWLMMSE filter is how to define the weight of each pixel. Inspired by the NL-means proposed in [21], the weight in our study is obtained by comparing the equality between the corresponding neighborhoods of the sample pixels and the processed pixel, which can yield a more robust comparison.

### 3.1. Calculation of Weights

In NL-means, the equality of two pixels is measured by the similarity of their neighborhoods. In [21], the authors employ the likelihood-ratio test of equality between two $W\left(\Sigma ,L\right)$ laws with the same number of looks under the complex Wishart distribution, as presented by Conradsen et al. [29], to calculate the equality between image patches.

By letting the independent Hermitian positive definite matrices X and Y be complex Wishart distributed, and considering the null hypothesis H_0_: $\Sigma_{x}$=$\Sigma_{y}$, which states that two matrices are equal, against the alternative hypothesis H_1_: $\Sigma_{x}\neq \Sigma_{y}$, the likelihood-ratio test statistic can be derived as follows:(13)$$Q=L(2p\ln2+\ln|X\left|+\ln|Y\right|-2\ln|X+Y|)$$
where *L* is the number of looks. The value of *Q* is nonpositive, and equals to 0 when two matrices are the same. For two image patches that each have K pixels, their equality E is regarded as the sum of the equalities of the corresponding pixels within the patch:(14)$$E=Q_{1}+Q_{2}+\cdots +Q_{k}$$

When filtering the pixel $p_{i}$, the equality $E\left(i,j\right)$ can be converted into the initial weight assigned to the sample $p_{j}$ by
(15)$$w_{j}=\exp(-\frac{E\left(i,j\right)}{h})$$
where the parameter h acts as the degree of filtering, and it controls the decay of the exponential function. In general, a low value of parameter h would lower the denoising performance of the NL-means and yield a noisy result. On the other hand, a high value of h would result in a burring denoised image. Figure 1 shows the profile of Equation (16) for a one-look image under a given h. We assume that, for an unknown single-look image, the equality E between two arbitrary patches randomly distributes between 0 and −100 (for E values of less than −100, the corresponding weights are much closer to zero, so they can be neglected). In NL-means, homogeneous pixels should occupy high weights when filtering a certain pixel. Therefore, a reasonable principle for choosing h is to ensure that the sum of the weights of the homogeneous pixels has a high proportion, such as 90% in this paper (black area of Figure 1). Based on the above assumption, the steps of estimating parameter h in this paper are summarized as follows: first, select the homogeneous areas in a single-look image, calculate the lnQ between any pair of pixels, and regard the mean value T of all the lnQ values as the boundary between the homogeneous pixels and heterogeneous pixels; then, choose a value of h to meet the condition that the integral from T∗K (K is the number of pixels in a patch) to 0 is equal to 90% of the integral from −100 to 0. By doing so, we obtain the approximate value of h as −3∗K. Since *Q* (Equation (13)) is proportional to the number of looks, the above theoretical value can be generalized to L look images as follows:(16)h=−3∗K∗L.

In this paper, the size of the neighboring patch is set as 3 × 3 pixels, thus K=9. Suppose we have N samples, then the normalized weight of a certain sample $p_{j}$ is computed as follows:(17)$$W_{j}=\frac{w_{j}}{\sum_{n=1}^{N}w_{n}}.$$

.

### 3.2. The Preliminary Step of Selecting Samples

Another question for our proposed NWLMMSE filter is how to select the samples. A common approach to NL-means is selecting all the pixels in a large square searching window, which leads to a high computational complexity, and makes it unpractical. Furthermore, strong point targets cannot be well retained by doing so, and the polarimetric scattering mechanism is not considered in the above equality calculation. The basic idea of the step proposed in this part is to preclassify the pixels in the searching window and to reject unrelated pixels according to their characteristics, with the aim of solving the above problems. This is performed as a preliminary step, and only those pixels that have similar characteristics to the processed pixel are selected as samples. This step is much faster than the weight calculation step.

In this study, two characteristics are considered to determine whether a pixel can be selected as a sample: scene heterogeneity (i.e., located in homogeneous scenes or heterogeneous scenes) and the polarimetric scattering mechanism.

To quickly obtain the scene heterogeneity of a pixel, we undertake a simple step to test the textural information of its neighboring patch, which is inspired by [30]. Three classes of scene heterogeneity are presented: homogeneous class, heterogeneous class, and isolated point target. The definition of the scene heterogeneity is derived from the coefficient of variation (CV) which is computed over the pixel’s 3 × 3 neighboring patch from the square roots of span. For a pixel, it is within a homogeneous area if $CV\leq C_{1}$; it is within a heterogeneous area if $C_{1}<CV<C_{2}$; and it is an isolated point target if $CV\geq C_{2}$. $C_{1}=0.523/\sqrt{L}$ and $C_{2}=\sqrt{1+2/L}$ are the theoretical thresholds provided and validated in [27], where L is the number of looks.

As mentioned in Section 2, apart from the statistical property, the polarimetric scattering mechanism should also be considered to measure the similarity of two pixels. To characterize the scattering mechanism of a pixel, we chose the scattering model-based decomposition by Freeman and Durden [31], which is based on realistic scattering models, for its effectiveness in providing scattering powers for each scattering component. In this polarimetric decomposition method, the pixels are categorized into three basic types: surface scattering dominated pixels, double-bounce scattering dominated pixels, and volume scattering dominated pixels. In this research, two pixels are considered to be similar in the polarimetric scattering mechanism, if they have the same dominant scattering mechanism.

In this study, isolated point targets are detected in the preliminary step and kept unfiltered. For a distributed target pixel being processed, only those pixels which have the same scene heterogeneity and dominant scattering mechanism in a square searching window are selected as the samples in the filter. For pixels located in homogeneous patches, the mean values of their neighboring patches can effectively represent these patches, so the equality of their neighborhoods can be obtained by comparing the equality of their mean values, to ease the computational load. 

The main purpose of this preliminary step is to preserve the strong scattering point targets and to consider the similarity of the scattering mechanisms between pixels, and in the meanwhile accelerate the algorithm. In fact, for a pair of pixels in a searching window, the similarity between them is calculated once when filtering each of them, which is repetitive and could be avoided by certain means in practice. A further discussion about the computational complexity of the algorithm is provided in Section 5. What must be made clear here is that the process of selecting pixels in this paper has a looser constraint than the conventional LLMMSE methods, which ensures that enough similar samples can be included in the filter.

## 4. Experiment Results of Polarimetric Speckle Filtering

In this section, to illustrate the performance of the NWLMMSE filtering method presented in Section 3, the results obtained with a simulated PolSAR image and two real-world polarimetric SAR images are reported. The proposed method is compared with the refined Lee filter with a 7×7 edge-aligned window, the NL-means method presented in [21], and the PolSAR nonlocal total variation-based filter (PolSAR NLTV) presented in [13]. In this paper, the searching windows of both NL-means and the NWLMMSE filter are set to be 17×17 pixels, and the size of the neighboring patches is 3×3 pixels.

### 4.1. Simulated Image

The use of simulated PolSAR data allows for an objective assessment of the performance of the proposed filter. Figure 2a shows the noise-free Pauli RGB image of the simulated data, and Figure 2b is the noisy generated image. The images not only contain linear and curvilinear edges but also contain some strong point targets. As we can see, the refined Lee filter reduces the speckle to some degree; however, the edges seem to be slightly blurred and the point targets are badly smeared. NL-means and NWLMMSE have comparable performances in both speckle reduction and edge preservation, but most point targets are lost in the image processed by NL-means. The reason for this is that in NL-means, when there are not enough pixels which are similar to the processed point target, the target will be blurred by unrelated pixels. The PolSAR NLTV filter shows better performance in retaining point targets than NL-means, but the boundaries of the objects seem to be slightly over-smoothed. The NWLMMSE filter can effectively preserve these strong point targets, since we have successfully detected them and retained their original values.

To further compare the performance in speckle reduction of each method, two quantitative indexes are introduced: the coefficient of variation (CV) and the equivalent number of looks (ENL). CV and ENL are two indexes that measure the smoothness of an image in the homogeneous areas. A low value of CV or a high value of ENL represents a low degree of speckle. Furthermore, to measure the reconstruction of edges, we also utilize the edge reconstruction error (ERR), as proposed in [32]. Table 1 lists the CV and the ENL values of each method, and it can be seen that that NL-means, NLTV, and NWLMMSE filters have much better performances in speckle reduction in homogeneous areas. For the ERR, the proposed method achieves the lowest value, since it preserves the details well and has a low error rate in reconstructing edges.

### 4.2. Real-World PolSAR Images

In this part, two real-world data sets, one from Flevoland in the Netherlands and the other from San Francisco, are used for demonstrating the performances of the proposed filter. Both of them are four-look.

Figure 3 displays the Pauli RGB image of the original data and the speckle filtering results of the different methods. The refined-Lee-filtered result (Figure 3b) reveals positive filtering characteristics; however, the scalloped appearance due to the edge-aligned windows in this filter may be undesirable for some applications, as pointed out in [8]. Compared with the refined Lee filter, NL-means and PolSAR NLTV (Figure 3c,d) show better performances in edge preservation, and the speckle is remarkably reduced in the processed images; however, the images of both filters seem to be somewhat oversmoothed, which leads to the problem that some details are slightly blurred. Furthermore, some isolated point targets are smeared as a kind of bright rectangle in the NL-means filtered image (Figure 3h). The result of the NWLMMSE filter preserves edges and point targets better, and the speckle is also reduced to a great extent.

An edge detector is often used as a tool to obtain the edge information of an image [33,34]. In this part, to qualitatively assess the edge preservation capability of the filters, the PolSAR constant false-alarm rate (CFAR) edge detector is used to detect the edge information of the images, since it is based on a test for equality of the covariance matrices and thus can utilize the full polarimetric information well [35]. The probability of false alarms for each image is set as −1.0, and the edge detection results are presented in Figure 4. As we can see, the CFAR detector has successfully detected most edges of the original speckled image in Figure 4a, but many “false edges” exist in homogeneous areas due to the influence of the speckles. The refined Lee filter (Figure 4b) has removed these false edges in its suppression of speckle; however, strong point targets and some fine edges are blurred and have not been detected by the detector (marked by the red ellipse). The NL-means and PolSAR NLTV filters (Figure 4c,d) have better performances than the refined Lee filter in the preservation of edges, but some point targets can still not be detected. The NWLMMSE filter reveals the best performance in edge preservation, and most of the point target signatures have also been retained.

To visualize the speckle reduction efficiency of the filters, we plot the scattergrams of the original and the filtered elements of the covariance matrix (C22 and the phase part of C13) for four land-cover types of the Flevoland image (Figure 5). The distributions of C22 (i.e., the intensity of the HV polarization) and the phase part of C13 (i.e., the HH and VV phase difference) become more concentrated for the filtered data than for the original data, and these four classes are much more separable in the data filtered by the NL-means, PolSAR NLTV, and NWLMMSE filters.

A good PolSAR speckle filtering method should possess filtering traits that not only reduce speckle and preserve edges, but also preserve the polarimetric scattering mechanisms of the image, because scattering mechanisms have always been considered as a class of important polarimetric information, and they are useful in land-use classification and scene interpretation. However, in the refined Lee filter and NL-means, the polarimetric scattering mechanism has not been taken into account in selecting homogeneous pixels or calculating the equality of two pixels. In our method, the scattering mechanism is chosen as one of the characteristics to select the samples. Two pixels are considered to be similar in the domain of a polarimetric scattering mechanism if they have the same dominant scattering mechanism. To validate the preservation of the scattering mechanism of the new method, we applied Freeman and Durden decomposition to a subset of the San Francisco image filtered by the different methods (Figure 6). This data set contains several classes of typical land objects that have different scattering mechanisms (for example, water pixels dominated by surface scattering, man-made structure pixels dominated by double-bounce scattering, and tree pixels dominated by volume scattering). The result of the 7 × 7 refined Lee filter (Figure 6b) shows the problem of a block effect, and some details are overly enhanced. NL-means (Figure 6c) has a better performance in speckle reduction and edge preservation than the refined Lee filter, as expected. However, both of the above two filters have smeared some strong signals, which returns from double-bounce dominated building pixels in the forest area and from the ships on the water. Visually, PolSAR NLTV performs a little better in retaining point targets than NL-means. For the NL-means and PolSAR NLTV filtered images, the speckle has been greatly reduced in the middle part of the urban area; however, some polarimetric information of the plants in this area has been lost. This is because the polarimetric scattering mechanism has not been taken into account by the two filters, and some piecemeal volume scattering pixels (i.e., plants) are filtered by too many double-bounce scattering pixels (i.e., buildings) in the large searching window. The proposed method reveals good filtering characteristics of preserving the dominant scattering properties, reducing speckle, and retaining the spatial resolution (Figure 6e).

### 4.3. Computational Complexity

Although NL-means has shown its outstanding performance in filtering, the high computational complexity still makes this method quite slow in practice. The high computational load is due to the cost of the equality calculation for all the pixels in a large window. Assuming the equality calculation between two pixels to be one step, and that the processed image has the size of M × N pixels, then the entire calculation scale of NL-means is $O\left(M\times N\times 17^{2}\times 3^{2}\right)$, which is obviously a heavy load. For the PolSAR NLTV filter, the processing time is not only dependent on the cost of the equality calculation between patches, but also dependent on the calculation procedure of minimizing the variational function. Therefore, the PolSAR NLTV filter is generally quite time consuming. In this paper, the processing time of the NWLMMSE filter is also mainly dependent on the cost of the equality calculation. As we take a preliminary step, which is much simpler and faster than the weight calculation, to select similar pixels and reject unrelated pixels in the searching window, our method has been accelerated greatly. In addition, for pixels located in homogeneous patches, the equality between their neighboring patches is obtained by calculating the equality of their mean values, which can speed up the weight calculation of pixels in homogeneous scenes. Furthermore, as we also make an effort to merge some of the repetitive computations, the computational load of our method is further eased.

Table 2 lists the processing times of each method. It can be seen that NL-means and PolSAR NLTV are quite time consuming, and the proposed method outperforms the above two filters to a large degree. In fact, we found that our method was at least twice as fast as NL-means in most cases, and the more complicated the image was, the faster the NWLMMSE filter was when compared to the NL-means. This is because, for images that have a lot of complicated scenes, the preliminary step can effectively reject unrelated pixels and discard the useless weight calculation. However, it should also be pointed out that the processing time of the proposed method is still much longer than the refined Lee filter, which is due to the inherent property of the NL-means-based methods. More research needs be done in our future study to further reduce the processing time.

## 5. Conclusions

The conventional LLMMSE filters have the shortcomings of the limited selection range of pixels, comparing pixels only by their own characteristics, and the assumption of absolute homogeneity of the pixels. In this paper, a nonlocal weighted LMMSE filter was proposed for the speckle reduction of PolSAR data. For the processed pixel, an effective preliminary step was developed to select similar samples, and the idea of NL-means was employed to calculate the weight of each sample. A weighted linear minimum mean-squared error estimator is was derived to filter the pixels. Experiments were conducted on a single-look simulated image and two four-look AIRSAR images, revealing the good filtering performance of our method in reducing speckle, retaining edges and targets, and preserving the polarimetric scattering mechanism. Moreover, the proposed method has relatively high computational efficiency with regard to most nonlocal filters, which makes it more applicable in practice.

It should be noted that, as for the PolSAR imaging problem in practical application, motion compensation is another important issue. The motion errors will make the imaging results blurred. As pointed out in [36,37], speckle is independent of either well-focused stationary scenes or blurred moving targets. In theory, the motion errors can degrade the performance of the proposed despeckling method, which, however, has not been deeply investigated in this study. As far as we know, up to now, very few studies have investigated the issue of suppressing the speckle noise and compensating the moving errors in fully polarimetric SAR images at the same time, which will be a meaningful work in our future study.

## Figures and Tables

**Figure 1 sensors-21-07393-f001:**
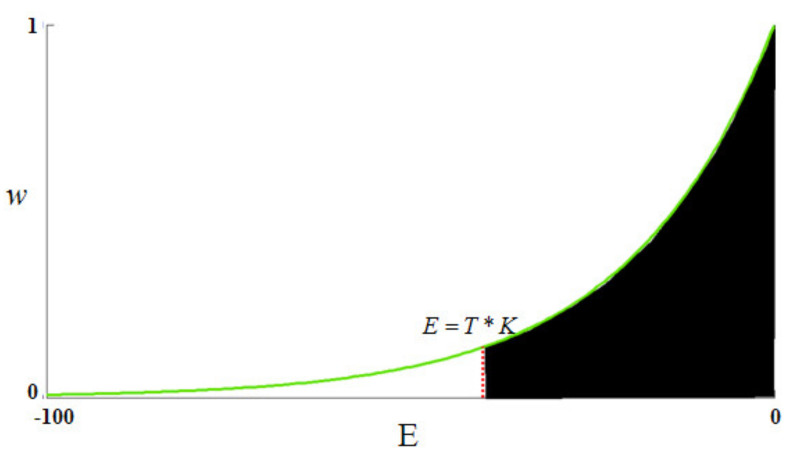
The profile of Equation (16) for a given h.

**Figure 2 sensors-21-07393-f002:**
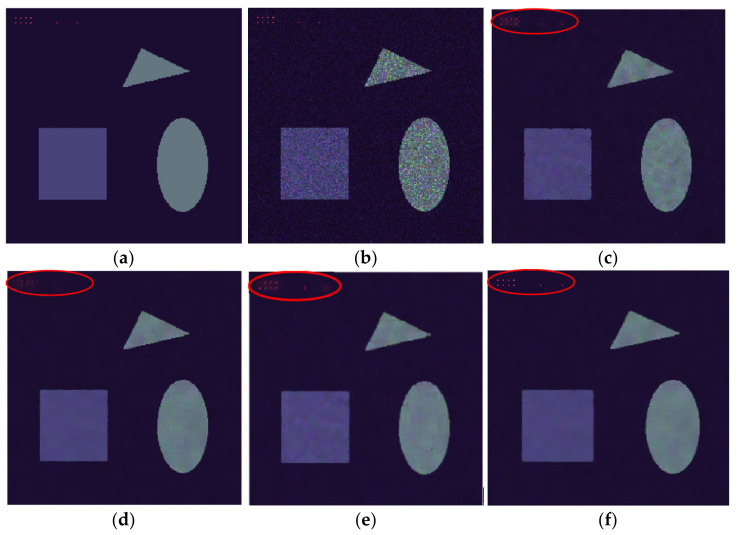
A comparison of different filters using a simulated image: (**a**) Pauli RGB image of the noise-free data; (**b**) single-look speckled image; (**c**) filtering result of the 7 × 7 refined Lee filter; (**d**) filtering result of the NL-means; (**e**) filtering result of PolSAR NLTV; and (**f**) filtering result of NWLMMSE filter.

**Figure 3 sensors-21-07393-f003:**
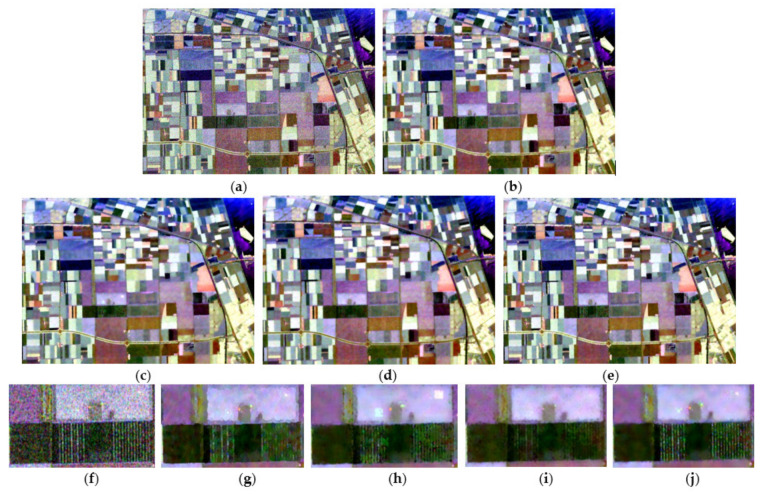
The filtering results for the Flevoland data set: (**a**) Pauli RGB image of the original data; (**b**) the result of the refined Lee filter shows a scalloped appearance in the image; (**c**) the result of NL-means has a remarkable performance in speckle reduction, but some details are overly smoothed and isolated point targets are smeared; (**d**) the image filtered by the NLTV shows the over-smoothing problem; (**e**) the image filtered by the proposed method is good with regard to both reduced speckle and retained details; and (**f**–**j**) detailed regions cropped from (**a**–**e**).

**Figure 4 sensors-21-07393-f004:**
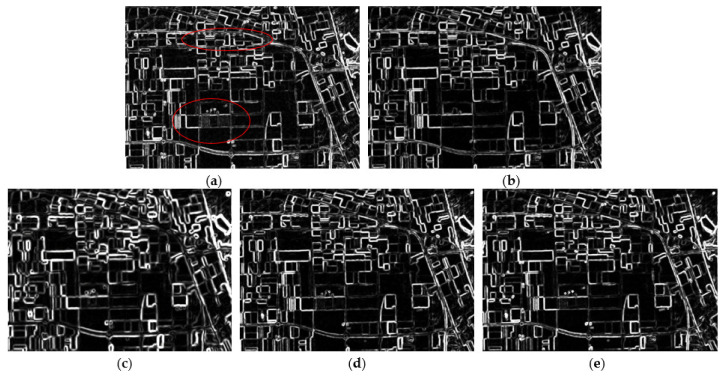
The edge maps obtained by the CFAR detector: (**a**) the edge map of the speckled image shows many “false edges” in homogeneous areas; (**b**) the edge map of the refined-Lee-filtered data, where false edges are removed, but some fine edges and point targets are missing; (**c**) the edge map of the NL-means-filtered data, where edges are retained well, but some point targets are still not retained; (**d**) the edge map of the NL-means-filtered data, where some point targets are still not retained; and (**e**) the edge map of the NWLMMSE-filtered image shows a better result in retaining edges and targets.

**Figure 5 sensors-21-07393-f005:**
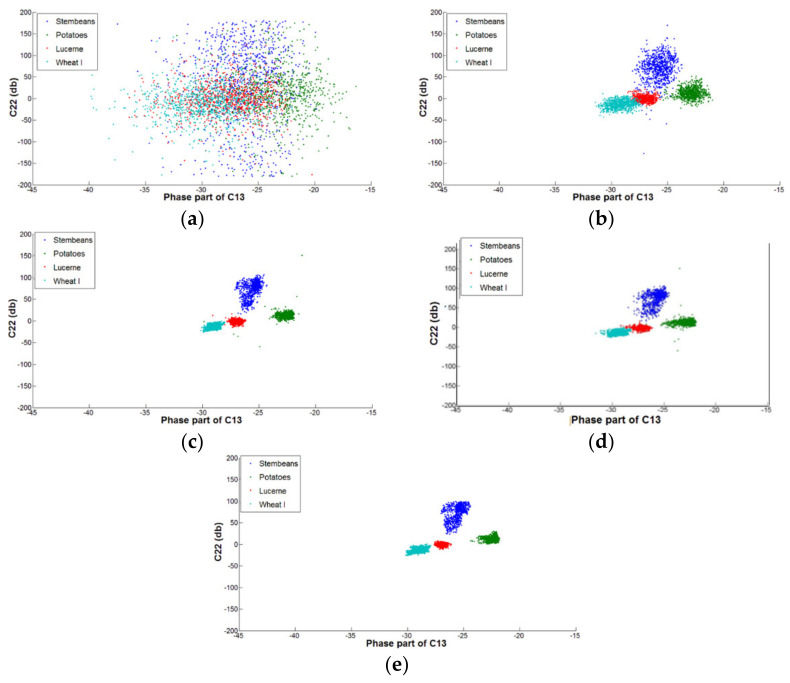
Distribution of the elements of the covariance matrix: (**a**) the distribution of C22 and the phase part of C13 for the original data and (**b**–**e**) the distribution of C22 and the phase part of C13 for the data filtered by the refined Lee filter, NL-means, PolSAR NLTV, and NWLMMSE filter, respectively.

**Figure 6 sensors-21-07393-f006:**
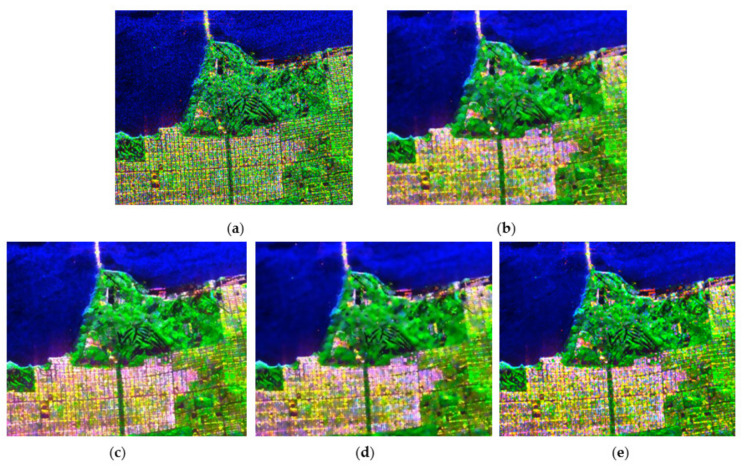
A comparison of the filtering results by Freeman and Durden decomposition to show their capabilities of preserving scattering properties: (**a**) Freeman RGB image of the original data; (**b**) the result of a 7 × 7 refined Lee filter, which brings in the problem of a block effect; (**c**,**d**) the results of NL-means and PolSAR NLTV show better performance in speckle reduction than the refined Lee filter, but the scattering mechanisms in urban areas and point targets are not well preserved; and (**e**) the result of our method effectively preserves the dominant scattering properties, reduces speckle, and retains the spatial resolution.

**Table 1 sensors-21-07393-t001:** Performance comparison for the different methods.

	Refined Lee Filter	NL-Means	PolSAR NLTV	NWLMMSE Filter
Mean value of CV	0.073	0.058	0.063	0.054
ENL	39.9	81.3	79.5	93.8
ERR	3.66	1.63	1.66	1.50

**Table 2 sensors-21-07393-t002:** Processing time (in seconds) of the different methods.

	Refined Lee Filter	NL-Means	PolSAR NLTV	NWLMMSE Filter
Simulated image (250 × 250 pixels)	21	608	1114	302
Flevoland (1024 × 750 pixels)	98	2203	3955	808
San Francisco (501 × 421 pixels)	40	1177	1898	402

## Data Availability

The data presented in this study are openly available in https://earth.esa.int/eogateway/ (accessed on 4 November 2021).

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
