# Peer review of "Polarimetric SAR Speckle Filtering Using a Nonlocal Weighted LMMSE Filter"

_sensors, 2021, doi:10.3390/s21217393_

Round 1

Reviewer 1 Report

The author proposed a nonlocal weighted linear minimum mean-squared error 11 (NWLMMSE) filter for PolSAR speckle filtering. A simulated image and two real-world PolSAR images are used to verify the effectiveness of the proposed algorithm. The total idea is interesting. I have the following comments:

  1. How about the computational cost of the proposed algorithm?
  2. Please compare with the-state-of-the-art. There are many novel approaches[1-3] for PolSAR image de-speckle. Please cite them and compare the performance with them.
  3. As for PolSAR imaging problem in practical application, motion compensation is another important issue. Is there any influence on the performance of the proposed approach if there are some motion errors. The motion error will make the imaging results blurred[4]. Is the proposed algorithm capable of dealing with the blurred area[5]? It’s may be intractable to consider motion errors in the proposed algorithm and experiments, however, some discussion on this point would be useful if tractable.

[1] Y. Ren, J. Yang, L. Zhao, P. Li and L. Shi, "SIRV-Based High-Resolution PolSAR Image Speckle Suppression via Dual-Domain Filtering," in IEEE Transactions on Geoscience and Remote Sensing, vol. 57, no. 8, pp. 5923-5938, Aug. 2019

[2] D. Tucker and L. C. Potter, "Speckle Suppression in Multi-Channel Coherent Imaging: A Tractable Bayesian Approach," in IEEE Transactions on Computational Imaging, vol. 6, pp. 1429-1439, 2020

[3] S. -W. Chen, X. -S. Wang and S. -P. Xiao, "Polarimetric SAR Speckle Filtering Based on Similarity Test and Adaptive Clustering," in IEEE Geoscience and Remote Sensing Letters, vol. 18, no. 4, pp. 702-706, April 2021

[4] W. Pu, "Deep SAR imaging and motion compensation," IEEE Trans. image process., vol. 30, pp. 2232-2247 2021.

[5] W. Pu, "Shue GAN With Autoencoder: A Deep Learning Approach to Separate Moving and Stationary Targets in SAR Imagery," IEEE Trans. Neural Networks and Learning Systems, 2021.

Author Response

We sincerely thank the reviewer for their very constructive and helpful comments and suggestions during the whole review process. We have made corrections or changes as the reviewers suggested, and we are now resubmitting a revised manuscript which we hope will meet with your approval. The major revised portions are marked in green in the revised manuscript. The item-by-item responses to the reviewer’s comments are listed as follows:

Comment 1: How about the computational cost of the proposed algorithm?

Response: Thanks for your comment. In fact, in the original paper, we have discussed the computational cost of the proposed method in Section 5 (now we put it in the end of Section 4) and listed the computational time in Table 2.

Comment 2: 2.    Please compare with the-state-of-the-art. There are many novel approaches for PolSAR image despeckle. Please cite them and compare the performance with them.

Response: This is a good suggestion. We have cited more new references in the introduction section of the revised paper. We have also compared our method with the PolSAR nonlocal total variation filter (see reference [22]). This state-of-the-art method also employ the idea of nonlocal means as our method and the theory of variational regularization, hence it is fair to directly compare it with the proposed method; besides, the source codes of this method are open available, so we can easily conduct the comparison experiments.

Comment 3: As for PolSAR imaging problem in practical application, motion compensation is another important issue. Is there any influence on the performance of the proposed approach if there are some motion errors. The motion error will make the imaging results blurred. Is the proposed algorithm capable of dealing with the blurred area? It’s may be intractable to consider motion errors in the proposed algorithm and experiments, however, some discussion on this point would be useful if tractable.

Response: We think these are very good questions. In the revised paper, we have explained and discussed the issue of motion errors in the end of the conclusion section: “It should be noted that, as for PolSAR imaging problem in practical application, motion compensation is another important issue. The motion errors will make the imaging results blurred. As pointed out in [37] and [38], speckle is independent of either well-focused stationary scene or blurred moving targets. In theory, the motion errors can degrade the performance of the proposed despeckling method, which, however, has not been deeply investigated in this study. As far as we know, up to now, very few studies have investigated the issue of suppressing the speckle noise and compensating the moving errors in fully polarimetric SAR images at the same time, which will be a meaningful work in our future study.”

Reviewer 2 Report

The authors address a very current and important issue, such as the development of speckle filtering for applications that use PolSAR data, they solve it in an elegant, operational and useful way (NWLMMSE), pre-classifying the pixels that allows to reduce stains, conserve the original singularities, the edges and the polarimetric dispersion mechanism, solving the problem of the heterogeneity of the pixels and the difficulty of applying them to large windows.

I consider that section 5. Discussion includes only a small part of the discussion of the work and includes results that should go in the results section, I suggest that the results and the discussion be included in the same section, or, where appropriate, expand the section 5 with a broader discussion, always avoiding redundancies, and put the table in the results section.

The contents included in the conclusions section correspond more to results and in any case to a description of the procedure rather than to draw conclusions from the work carried out, perhaps this section could be improved.

Author Response

We sincerely thank the reviewers for their very constructive and helpful comments and suggestions during the whole review process. We have made corrections or changes as the reviewers suggested, and we are now resubmitting a revised manuscript which we hope will meet with your approval. The major revised portions are marked in green in the revised manuscript. The item-by-item responses to the reviewer’ comments are listed as follows:

Comment 1: I consider that section 5. Discussion includes only a small part of the discussion of the work and includes results that should go in the results section, I suggest that the results and the discussion be included in the same section, or, where appropriate, expand the section 5 with a broader discussion, always avoiding redundancies, and put the table in the results section.

Response: Thanks for your constructive suggestion. In the revised paper, we have put Section 5 of the original manuscript to the end of Section 4. Besides, as Reviewer #1 suggested, we have compared the proposed method with a new state-of-the-art filter in Section 4, and more contents have been added in Section 4.

Comment 2: The contents included in the conclusions section correspond more to results and in any case to a description of the procedure rather than to draw conclusions from the work carried out, perhaps this section could be improved.

Response: We think this is a good comment. We have improved the conclusion section in the revised paper. Firstly, the issue of computational efficiency of the proposed method is mentioned; secondly, we have explained and discussed the issue of motion errors in the end of the conclusion section: “It should be noted that, as for PolSAR imaging problem in practical application, motion compensation is another important issue. The motion errors will make the imaging results blurred. As pointed out in [37] and [38], speckle is independent of either well-focused stationary scene or blurred moving targets. In theory, the motion errors can degrade the performance of the proposed despeckling method, which, however, has not been deeply investigated in this study. As far as we know, up to now, very few studies have investigated the issue of suppressing the speckle noise and compensating the moving errors in fully polarimetric SAR images at the same time, which will be a meaningful work in our future study.”